# Fluid network dynamics in the prefrontal cortex during multiple strategy switching

Hugo Malagon-Vina[1], Stephane Ciocchi[1,3], Johannes Passecker [1], Georg Dorffner[2] & Thomas Klausberger[1]

Coordinated shifts of neuronal activity in the prefrontal cortex are associated with strategy adaptations in behavioural tasks, when animals switch from following one rule to another. However, network dynamics related to multiple-rule changes are scarcely known. We show how firing rates of individual neurons in the prelimbic and cingulate cortex correlate with the performance of rats trained to change their navigation multiple times according to allocentric and egocentric strategies. The concerted population activity exhibits a stable firing during the performance of one rule but shifted to another neuronal firing state when a new rule is learnt. Interestingly, when the same rule is presented a second time within the same session, neuronal firing does not revert back to the original neuronal firing state, but a new activity-state is formed. Our data indicate that neuronal firing of prefrontal cortical neurons represents changes in strategy and task-performance rather than specific strategies or rules.

[1] Center for Brain Research, Divison of Cognitive Neurobiology, Medical University of Vienna, 1090 Vienna, Austria. [2] Center for Medical Statistics, Section for Artificial Intelligence, Informatics and Intelligent Systems, Medical University of Vienna, 1090 Vienna, Austria. [3]Present address: Department of Physiology, University of Bern, 3012 Bern, Switzerland. Correspondence and requests for materials should be addressed to H.M.-V. (email: hugo.malagonvina@meduniwien.ac.at) or to T.K. (email: thomas.klausberger@meduniwien.ac.at)

In ancient Greek, Heraclitus famously stated that "No man ever steps in the same river twice, for it's not the same river and he's not the same man". He referred to the ambiguity that conscious actions and plans are never truly experienced the same way, however similar they may appear. Earlier research demonstrated how distinct behavioural rules and strategies are entailed in the activity of prefrontal neurons[1–3]. We aim to address the question how a certain behavioural strategy, applied on two different occasions within the same session, is represented in the neuronal firing rate of the prefrontal cortex. The prefrontal cortex is a central structure for executive control of flexible behaviour to assess new rules and strategies, not only in humans[4,5] but also in monkeys[6,7] and rodents[8–10], and is highly interconnected with other brain regions[11,12] indicative of an integrative structure[13] and a multifunctional role during cognitive tasks[14]. The importance of the prefrontal cortex is highlighted by neuronal firing patterns contributing to error-related activity[15–17], working memory[18–23], decision making[24–28], and reward encoding[29–33]. It has been shown that lesions of the medial prefrontal cortex lead to an impairment of the ability to follow changing spatial rules[34]. Prefrontal neurons also change their firing activity when animals are switching between different strategies[1]. This indicates the importance of the prefrontal cortex for rule-guided behaviour. In addition, behavioural rule changes can lead to abrupt neuronal population activity changes within a very short period of time[2]. Furthermore, abrupt and coordinated changes of neuronal firing have been reported when animal behaviour reflects uncertainty and evaluation of possible new strategies[3]. However, the following questions remain unclear: (1) how multiple and consecutive changes of strategy would be reflected in the firing of prefrontal neurons and (2) how the same repeated strategy would be represented on different occasions.

To address neuronal computations in consecutive rule presentations, we used single-unit recordings in freely moving rats to assess neuronal activity during a prefrontal cortex-dependent rule-switching task[10]. In our task design, animals managed to perform under multiple rules during a single recording session, which allowed us to study the neuronal representations when animals changed new rules and acquired new strategies. Additionally, we examined how neuronal population states are changing during the presentation and repetition of multiple rules in the course of a single session.

The results of this study show that the neuronal population forms a different and stable firing-state every time a new rule is learnt, even when the same rule is presented twice during the same session. This implies that the concerted neuronal population in the prelimbic cortex does not represent individual rules permanently, but it reflects a change of strategies.

## Results

**A strategy-switching task with multiple rule changes.** Rats ($n = 5$) performed a strategy-switching task with multiple rule changes (from 1 up to 6 changes, median = 3) within each behavioural session (Fig. 1a), while the activity of neurons in the prefrontal cortex was extracellularly recorded with 12 tetrodes (Fig. 1b). Rats were seeking a food reward on a plus-maze using one out of four possible strategies based on two allocentric (landmark-referenced) and two egocentric (self-referenced) rules. The animals were placed at one of the two possible starting arms (North or South arm) and they had to decide to run towards one of the two goal arms (East or West arm), while the arm opposite of the starting position was blocked. After reaching the end of the goal arm, a reward was given for a correct choice according to the current rule. Then the animal was manually positioned into a bin at the centre of the maze to break stereotyped behaviour. After

3–7 s, the rat was placed again at one of the starting arms to begin the next trial. When an animal successfully succeeded in performing 13 out 15 consecutive trials, the rule was changed without notice and the animal had to switch strategy in order to maximise reward based on trial and error information. We analysed the performance of the animal using the behavioural choices (correct and incorrect) using a Markov-chain Monte–Carlo analysis[35] (Fig. 1c), which defines the probability of the rat being correct during each trial together with the associated confidence intervals. Those intervals are used to determine learning periods (see Methods). Only one recording session was carried out on a single day.

We performed behavioural control experiments in two rats to assess whether animals indeed used landmarks during allocentric but not egocentric strategies. The maze was surrounded by four walls, which displayed distinct and large landmarks for the orientation of the rats during task navigation. The availability of landmark cues during allocentric and egocentric strategies was controlled by keeping or removing all landmarks for 10 trials, following the successful learning of a rule. When animals were allowed to use the visual landmarks during the allocentric rules, they performed significantly better than without the availability of landmarks ($p = 0.0004$). However, during egocentric rules, the removal or addition of landmarks did not alter performance ($p = 0.53$) (Fig. 1d). This indicates that rats, indeed, followed allocentric or egocentric strategies during the respective rules to maximise their reward.

**Neuronal firing correlates with task-performance.** A total of 300 neurons were recorded with tetrodes in the prefrontal cortex in three animals during the performance of the strategy-switching task. For the recording sessions, we determined the trial-by-trial time-series of neuronal firing rates for the entire trial, by diving the number of spikes of a neuron by the time of the trial. To test the consistency of our findings, we divided each trial into 3 non-overlapping behavioural trial segments defined as run, reward and inter-trial period (see Methods). By examining the firing rate of individual neurons during consecutive trials and different behavioural segments, we observed the following two groups of neurons (Fig. 2a, b): (1) positively correlated neurons that had an increased firing rate during behavioural periods with good performance, and (2) negatively correlated neurons, which had an increased firing rate during periods with low performance, when negative feedback in form of repeated lack of reward with a conflicting understanding of the task was experienced. Out of 300 recorded units, we identified neurons ($n = 95, 54, 74$ and 84 for the entire trial, run, reward and inter-trial period, respectively) with either significant negative or positive correlations between their firing rate and task performance (Fig. 2c, Spearman's correlation, $\alpha = 0.05$, Bonferroni–Holm correction; the data for different animals are shown in Supplementary Fig. 1). The cumulative distribution function of the correlations ($n = 300$ neurons) in different trial segments (Fig. 2d) indicate that subsets of prefrontal neurons exhibit either a positively or negatively correlation of firing rate with task performance of the animal. To demonstrate that the correlations observed between firing rate and task performance are rather a product of the integration of outcomes over time rather than reflecting an instantaneously received reward or lack thereof, we shuffled the order of trials while keeping the firing rate and correct or incorrect performance associated with each trial, and generate a new shuffled performance curve. Then, we tested for possible correlations between firing rate and shuffled performance and observed that the correlations obtained from shuffled trials were significantly lower than those derived from observed data for all episodes as well as

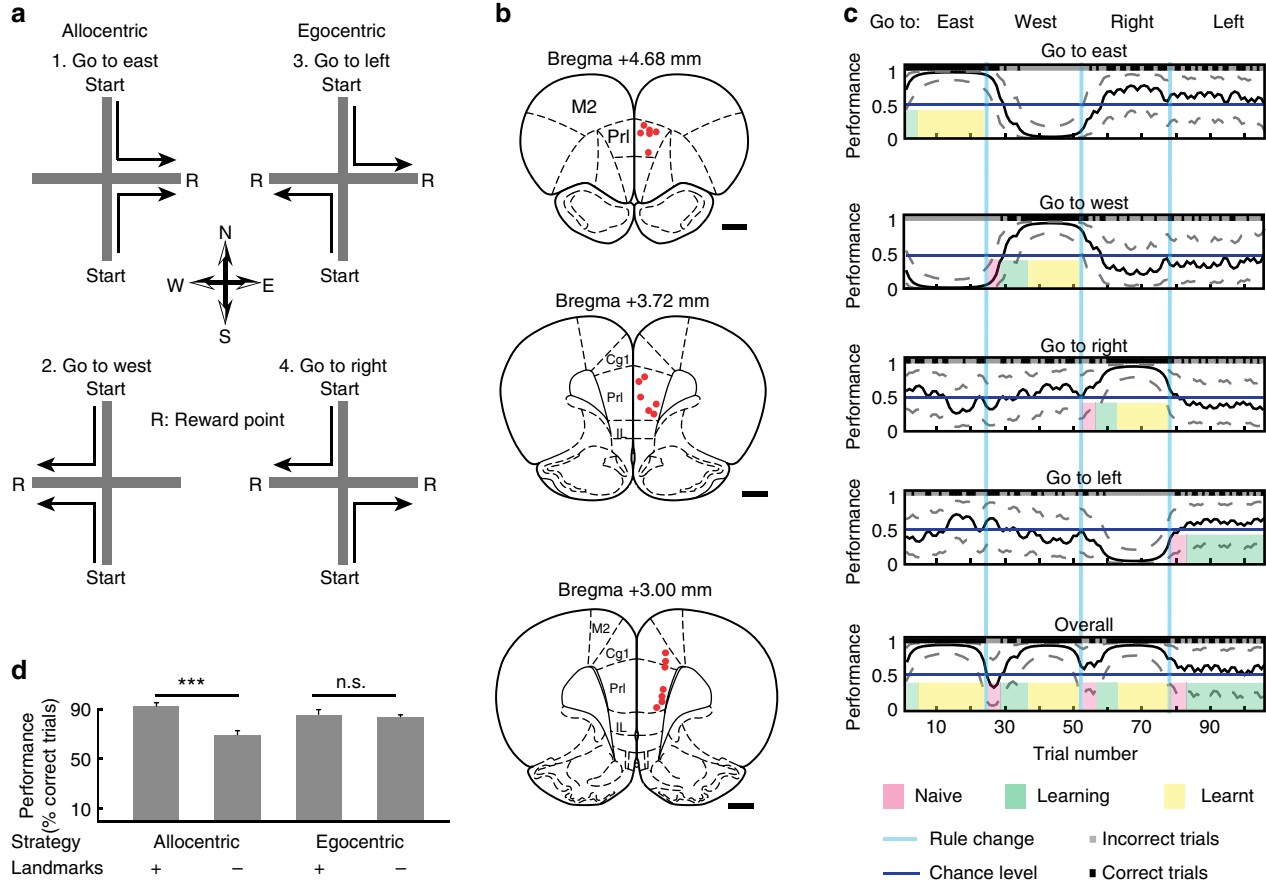

**Fig. 1** Strategy-switching task and behavioural analysis. **a** Rats were placed randomly at one of the two possible start positions during consecutive trials. On the basis of landmark-referenced (allocentric strategy, rules: 1, 2) or self-referenced (egocentric strategy, rules: 3, 4) navigation, the animal has to travel on a plus-maze in order to receive reward (R). After the rat makes 13 correct choices within 15 consecutive trials, the rule is changed unannounced to the animal. **b** Position of the recording sites (n = 19, number of rats = 3) in the prelimbic and cingulate cortex indicated by red dots in three different coronal sections of the prefrontal cortex. Scale bars, 1 mm. **c** Task performance of a behavioural session was evaluated using the binary data of behavioural choices (correct choice: black, incorrect choice: grey) via a Markov-chain Monte–Carlo analysis[35] which provides a confidence interval for each trial. By thresholding the performance score of the lowest confidence interval in any of the learning curves (go east, go west, go to right or go to left) corresponding to different rules, three different behavioural phases were assigned to each trial: naive (below 0.1 for reversals and 0.3 for switches); learning (between 0.1 and 0.6 for reversals and 0.3–0.6 for switches); and learnt (over 0.6). **d** Task-performance during control experiments when landmarks on surrounding walls were removed or maintained during allocentric (n = 11) or egocentric (n = 13) strategies. Keeping the landmarks during allocentric strategies (n = 12 tests; performance = 90 ± 3.015 %; data are mean ± SEM), removing the landmarks during allocentric strategies (n = 11, performance = 68 ± 3.71 %), keeping the landmarks during egocentric strategies (n = 12, performance = 83 ± 4.14 %) and removing the landmarks during egocentric strategies (n = 13, performance = 81 ± 1.91%). Note that only the removal of landmarks during allocentric strategies resulted in a reduced task performance (Wilcoxon rank-sum test). *** indicates p < 0.001, N.S. indicates no significant. Error bars, s.e.m

for the entire trial time (Fig. 2e, difference tested with Kolmogorov–Smirnov test). This indicates that correlations between firing rates of neurons and performance of the animal correspond to the integration of previously experienced trials and not to the instantaneous response to the reward obtained in each trial.

As fast-spiking GABAergic interneurons might be important for computations involving negative feedback and conflicting results[36–38], we divided neurons into two groups: one group of neurons with firing rate higher or equal to 10 Hz, in which an enriched—but not exclusive—population of fast-spiking interneurons is expected, and another group of neurons with firing rates lower than 10 Hz, in which an enriched—but not exclusive—population of pyramidal cells is expected (Fig. 3a). When compared with low-firing neurons, a larger fraction of high-firing neurons had significantly correlated firing rates with task performance, (Fig. 3b, p = 0.024, chi-square test) and also

displayed an enriched tendency to be negatively correlated (Fig. 3b, p = 0.014, $X^2$ test).

**Prefrontal neurons reflect changes in rules and strategies.** The neuronal state representation in the prefrontal cortex changes during rule switches[2]. However, the neuronal state dynamics remain unclear when multiple rule changes are presented in the same recording session. As the activity of the prefrontal cells reflects behavioural performance, we tested changes in population activity during multiple rules presentations in a single recording session. For this, each trial was represented as a population vector of neuronal activity: $T = FR_1, FR_2, ..., FR_n$, where $FR_n$ is the firing rate of neuron 'n' during trial $T$. For this analysis, we only used trials during learning and learnt phases of the presented rules. We excluded naive phases because they consist of mainly persistent behaviour of the animal still following the previous rule (Fig. 4a). We observed that the N-dimensional population vectors of

neuronal activity belonging to one rule tended to be clustered, at least when dimensionality reduction using a principal component analyses was applied (Fig. 4b). To test this quantitatively and without dimensionality reduction, we first applied a K-means clustering algorithm (see Methods section) to the N-dimensional representation, specifying the number of clusters as the number

of rules that should be found. The algorithm assigned each of the trials to a specific "rule cluster". We compared this assignment with the actual rules during the respective trial and created an accuracy index (number of correctly grouped trials over the total number of trials). We tested the null hypothesis that population vectors, defined by the firing rates, do not cluster according to

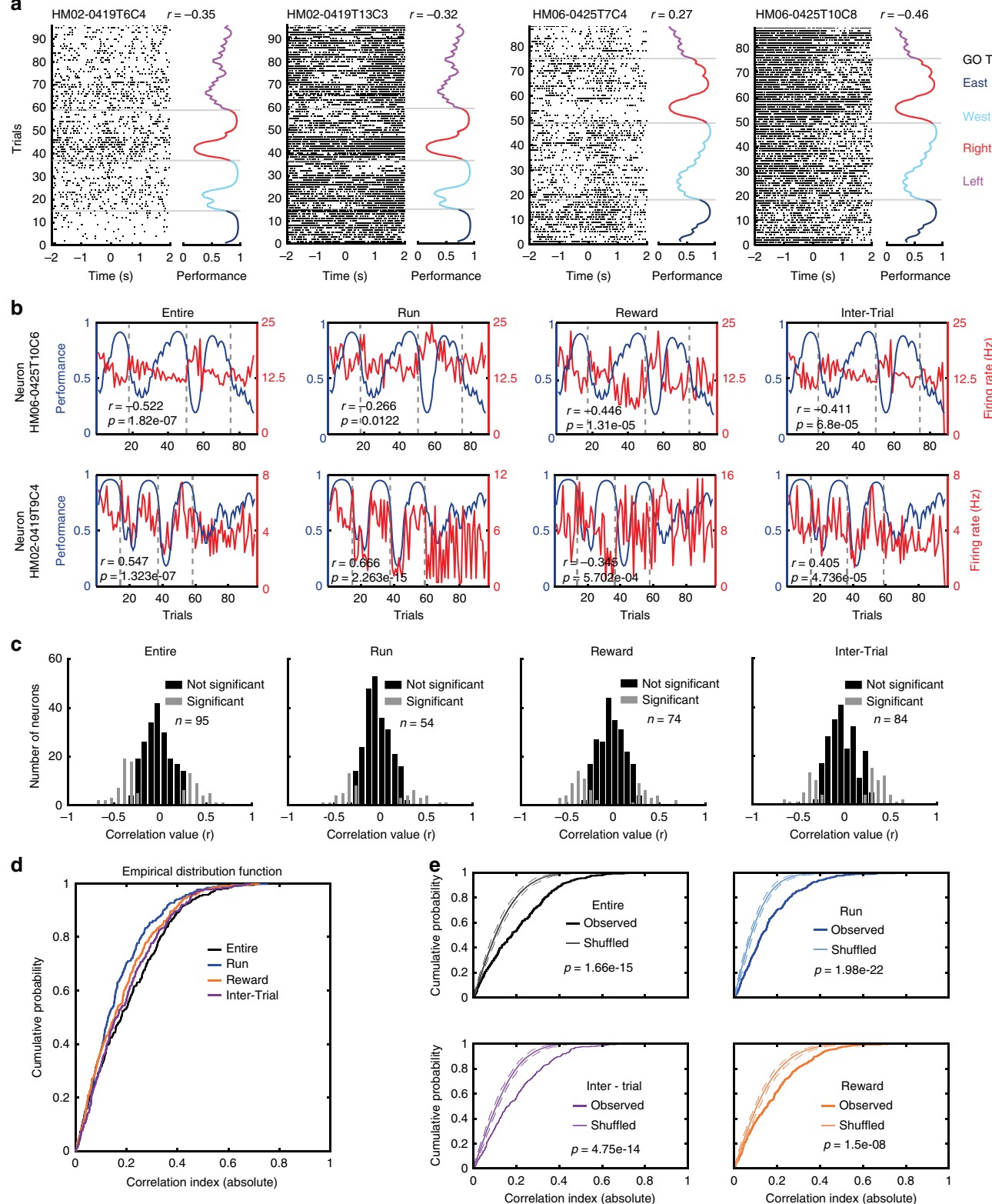

rules using a permutation test. We shuffled the order of trials keeping the firing rate associated with each trial, applied the K-means algorithm and recomputed the accuracy index. The accuracy of the K-means algorithm is significantly higher for the observed data than for the shuffled data, demonstrating that there is a clustered organisation of the trials (Fig. 4c, $p = 3e-5$, Wilcoxon signed-rank test and Supplementary Fig. 2a for data from individual animals). We could confirm that the centre of mass of the N-dimensional cloud belonging to a rule can be used as a representation of the clustered rule (Fig. 4d and Supplementary Fig. 2b). Clustering using Mahalanobis distance also produce a significant difference between observed and shuffled data (Supplementary Fig. 3a, $p = 1.07e-04$, Wilcoxon signed-rank test). We found a significant difference ($p = 1.8e-5$, Wilcoxon signed-rank test) between observed and shuffled data, when the clusters were defined by the centre of mass. Overall, due to the observation that N-dimensional population vectors of neuronal activity can be clustered, this implies that neuronal activity of the prefrontal cortex holds some form of a representation of rules.

Nonetheless, it has been shown previously that representations in the prefrontal cortex are drifting over time[39], which may account for the clustered organisation of our data across consecutive trials and rules. To address this possibility, we performed a multiple linear regression in which we explained the distance between the N-dimensional centres of two clustered rules either by the number of trials (as a measure of time) or by the number of rules separating them. The distance between two rules was calculated by measuring the Euclidean distance between the N-dimensional centres of mass of both clusters. The resulting plane of the regression showed a significant explanation of the distance due to the number of rules in between ($p = 2.81e-08$) but not due to the trials in between ($p = 0.78$, Fig. 4e). Supplementary Table 1 shows the p-values segregated per animal. The interaction between both factors was not significant ($p = 0.704$). To further elucidate the contribution of both time and rules to the distance between the rules, two partial correlations were calculated (Fig. 4f). The partial correlation of the distance between two rules and the number of trials in-between is not significant ($p = 0.779$, Spearman correlation) when the number of rules in-between is taken into account. On the contrary, the partial correlation of the distance of two rules and the number of rules in-between is significant ($p = 2.8e-8$, Spearman correlation), even when the number of trials in-between is taken into account. Moreover, when Mahalanobis distance was used instead of Euclidean, similar results were obtained. The Mahalanobis distance between clusters is significantly explained by the number of rules ($p = 4.756e-05$) but not by the number of trials in between the clusters ($p = 0.76405$). In addition, the interaction term is not significant ($p = 0.646$). The partial correlations, using Mahalanobis distances, follow the same tendency (Supplementary Fig. 3b, $p = 4.756e-05$ (top), $p = 0.764$ (bottom), Spearman correlation). Overall, this data indicates that the distance of firing vectors between rule clusters is better explained by the presentation of rules per se, rather than by time that has passed

between rules, suggesting that rule-dependent switches in strategy may drive the population activity rather than just the passing of time.

**Repeated rules do not induct the same firing state.** Having discovered a clustering of neural firing patterns in the prefrontal cortex based on rules, this leads to the question whether those firing patterns could reflect persistent representations of a specific rule. To address this question, we presented the animals with the same rule twice within a single recording session (Fig. 5a, b). Strikingly, after visualising the data projected onto the two first principal components, we observed that when the animal faced the same rule a second time later in the session, the neuronal population formed a new state instead of returning to the one initially formed when the rule was presented for the first time (Fig. 5c, d, e, f). To further corroborate that the neuronal population response of the repetition 'A°' of a given rule 'A' is distinct, we trained two different classifiers: a logistic regression and a support vector machine. The data were divided into a training set and a test set. The training set consisted of 70% of the trials corresponding to the rule 'A' and it was labelled as '1', and the trials of all other rules (except the repetition 'A°'), which were labelled as '0'. The test set included the repetition 'A°' and the remaining 30% of the trials in 'A'. After building the decoder with the training set, an accuracy value of belonging to the rule 'A' is computed for the test set (number of trials classified as '1' over the total number of trials). The classifiers correctly assigned the data belonging to rule 'A' as '1' (belonging to rule 'A'), while the data of rule 'A°' is assigned as '0', indicating a different rule form 'A' (Fig. 5g, SVM $-> p < 1e-20$, Logistic regression $-> p < 1e-20$). These analyses suggest that the neuronal firing state during a rule repetition is different from the firing during the initial rule.

Interestingly, both the Euclidean and Mahalanobis distance between two repeated rules are not significantly different from the distance between two different rule clusters with the same number of rules in between (Fig. 5h, $p = 0.58$, Wilcoxon rank-sum test and Supplementary Fig. 3c, $p = 0.248$, Wilcoxon rank-sum test), indicating that the repetition of the rule is perceived by the prefrontal cortex's neuronal population in a similar way as if another new rule was presented.

These results strongly advocate that cognitive rules are not specifically represented by unique neuronal firing in the prelimbic and cingulate cortex, but rather by a new formation of neuronal firing states, even during the repetition of rules during the same session.

**Speed and trajectory do not explain firing states.** It is known that firing in the prefrontal cortex neurons reflects animal trajectories and movement[40]. Thus, it might be possible that the different neuronal firing states during a rule and during its repetition might be due to the change in trajectories or running speed which might variate between the beginning of the recording session and the end. However, trajectories of the animal do not

---

**Fig. 2** The firing rate of individual neurons correlates with task-performance of the animal during multiple rule changes. **a** Individual spikes (black ticks) are plotted around the time of reward activation (0 s). The corresponding behavioural performance is plotted at the right of the raster plot and colour coded according to rules. Note that spike activity is modulated by the performance. The correlation value (r) between trial-by-trial firing rate and performance is indicated (Spearman correlation). Grey lines denote rule changes. **b** The firing rate of two neurons (Neuron HM06-0425T10C6 and Neuron HM02-0419T9C4) are negatively or positively correlated to performance during the task irrespective of the trial time segment. **c** Histograms of correlation values (firing rate versus task-performance) for 300 recorded neurons and for different trial segments. n represents number of neurons with firing rates significantly correlated to task performance (Spearman´s correlation, $p < 0.05$ after Bonferroni–Holm correction). **d** Cumulative distribution functions of individual neurons' correlation values for different trial segments. **e** Comparison of observed and shuffled cumulative distribution functions of individual neurons' correlation values for different trial segments. Dotted lines indicate confident intervals at 2.5 and 97.5 %. Note that the firing rates of some neurons are correlated with task performance during all task periods

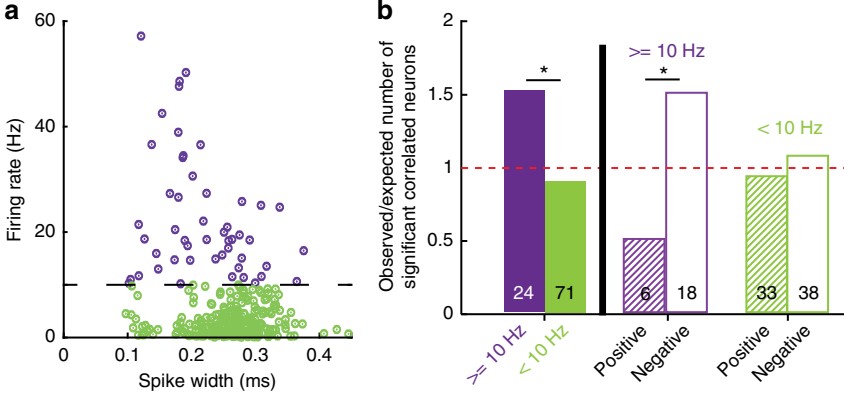

**Fig. 3** High-firing neurons are negatively correlated with task performance. **a** Scatter plot for firing rates versus spike width of all recorded neurons. Threshold for separation of high (purple) and low (green) firing neurons was arbitrarily set at 10 Hz. **b** Comparison of different groups for neurons with significant correlation between firing rate and task-performance of the animal. Y-axis indicates the ratio between observed number of significantly correlated neurons in a group and numbers expected by chance. Numbers at the bottom of each bar indicate n of each group. * indicates $p < 0.05$; $X^2$ test. " $\geq 10$ Hz" (purple) and " $< 10$ Hz" (green) indicate neurons with respective firing rates above or below 10 Hz

seem to variate much from one rule to its repetition (Fig. 6a, Supplementary Fig. 4). Nonetheless, in order to account for the trajectories and include them in our analyses, we decided to take advantage of the maze's geometry and fit the trajectories to a quadratic equation: $ax^2 + bx + c$. The three coefficients $a$, $b$, $c$ describe the trajectory of an animal in a given trial, and we could extend our analyses to account for movement (Fig. 6b). Differences between trajectories of trials were quantified by the three coefficients of the quadratic fit (Fig. 6c). We compared the distances of trials which have very similar trajectories (lower than 5th percentile of the similarity index distribution) between the first presentation of a rule 'A' and its repetition 'A°' vs the distances of trials, which have very different trajectories (higher than the 95th percentile of the similarity index distribution) within the rule 'A' (Fig. 6d). Even for trials with very similar trajectories between 'A' and 'A°', the Euclidean distances of firing states are significantly larger ($p = 0.0017$, Wilcoxon rank-sum test) than those between trials only within 'A' with very different trajectories. This implies that the trajectories per se do not explain the separated clustering of the rule repetition.

However, trajectories and speed still might have a more subtle effect. Therefore, we modelled the firing rate of a neuron by using the coefficients of the quadratic equation and the speed of the animal as follows: $FR_T = \beta_0 + a_T\beta_1 + b_T\beta_2 + c_T\beta_2 + s_T\beta_3$ where FR is the firing rate; $a$, $b$, $c$ are the coefficients of the quadratic equation and '$s$' is the speed in m/s in a given trial '$T$'.

As expected, close to a 30% of the neurons showed a significant correlation to at least one of the coefficients in each of the 4 possible paths (Supplementary Table 2a, b). After having a quantitative value of the trajectory contribution to the firing rate, we used the residuals of the model (the part of the firing rate that it is not explained by the movement variables) to re-do all the analyses presented in this paper. When using the residuals of the model, similar correlation values between the firing rate and the performance of the animal are maintained for all the neuronal population, as well as for the high firing rate neurons (Fig. 6e, f, $r = 0.9123$, $p < 1 \times 10^{-20}$ and $r = 0.9326$, $p < 1e-20$, Spearman correlation).

Moreover, projections of the residuals on the first two principal components of the multi-unit activity, still remain clustered (Fig. 6g) and rule repetitions are found to be still in another cluster different from the first presentation (Fig. 6h). The results of the accuracy of the K-means clustering algorithm applied to the residuals of the firing rates still show a clustered organisation of the rules in the network state (Fig. 6i). The same general linear

model previously described and shown in Fig. 4e was now fitted with the residuals data, reaching similar results. The number of rules in between is responsible for the explanation of the cluster's distance ($p = 0.00053$) and not the number of trials in between ($p = 0.97$). In addition, there is not a significant difference between the distances from data of non-repeated and repeated rules (Fig. 6j, $p = 0.3944$, Wilcoxon rank-sum test).

Overall, these results support that both the neuronal representations of cognitive rules and the new formation of a neuronal firing state upon the repetition of a rule are not an artefact of the movement of the animals.

## Discussion

To investigate the computations of neuronal populations in the prefrontal cortex during flexible behaviour, we recorded neuronal activity while animals were performing a prefrontal cortex-dependent strategy-switching task[1]. We managed to assess multiple rule changes within the same session, which allowed the possibility to study neuronal population dynamics while following several behavioural changes in strategy. We found two complementary neuronal groups, which dynamically changed their firing during behaviour and their firing rates were significantly correlated with performance. In addition, when multiple rules were presented on the same day, neuronal populations formed new neural states for each rule, even when the same rule was presented twice.

Neurons in the prefrontal cortex have a multitude of different firing patterns, which reflect many aspects of the external environment as well as internal computations, including goal-related firing[21,26,27,41], reward[29,31,32,42–44], encoding of memory and executive functions[18,20,41,45,46] or confidence[24]. But how do diverse firing patterns adapt when the strategy of an animal changes? Lesions[5,34,47,48], optogenetic inactivation[49] and pharmacological inactivation[10] of the prefrontal cortex have been reported to induce impairment in cognitive flexibility. In the prelimbic cortex, changes in firing rates of single neurons[1,50] and neuronal populations[2,51] have been observed in relation to flexibility in strategy during a rule switching task. Often those changes are presented as a global change of activity and might relate to a complete switch to a different network state, as shown for synchronised changes of neuronal populations during an uncertainty task[3]. In fact, after lesioning the medial prefrontal cortex, animals could not follow a change of spatial related rules[34].

We assessed how changes in network activity are related to the performance of the animal during multiple rule changes that are presented within the same session, reflecting a high demand on cognitive flexibility. In order to perform a goal-guided behaviour, often the right choice between stability and flexibility has to be found. Thus, the prefrontal network operations should be flexible enough to take into account different sensory inputs and experiences, which provide evidence for a different and more successful strategy, but at the same time should remain stable enough to ignore irrelevant information[52]. This may contribute to the reason why neuronal ensembles in the prefrontal cortex often present abrupt changes in activity whenever a new strategy is

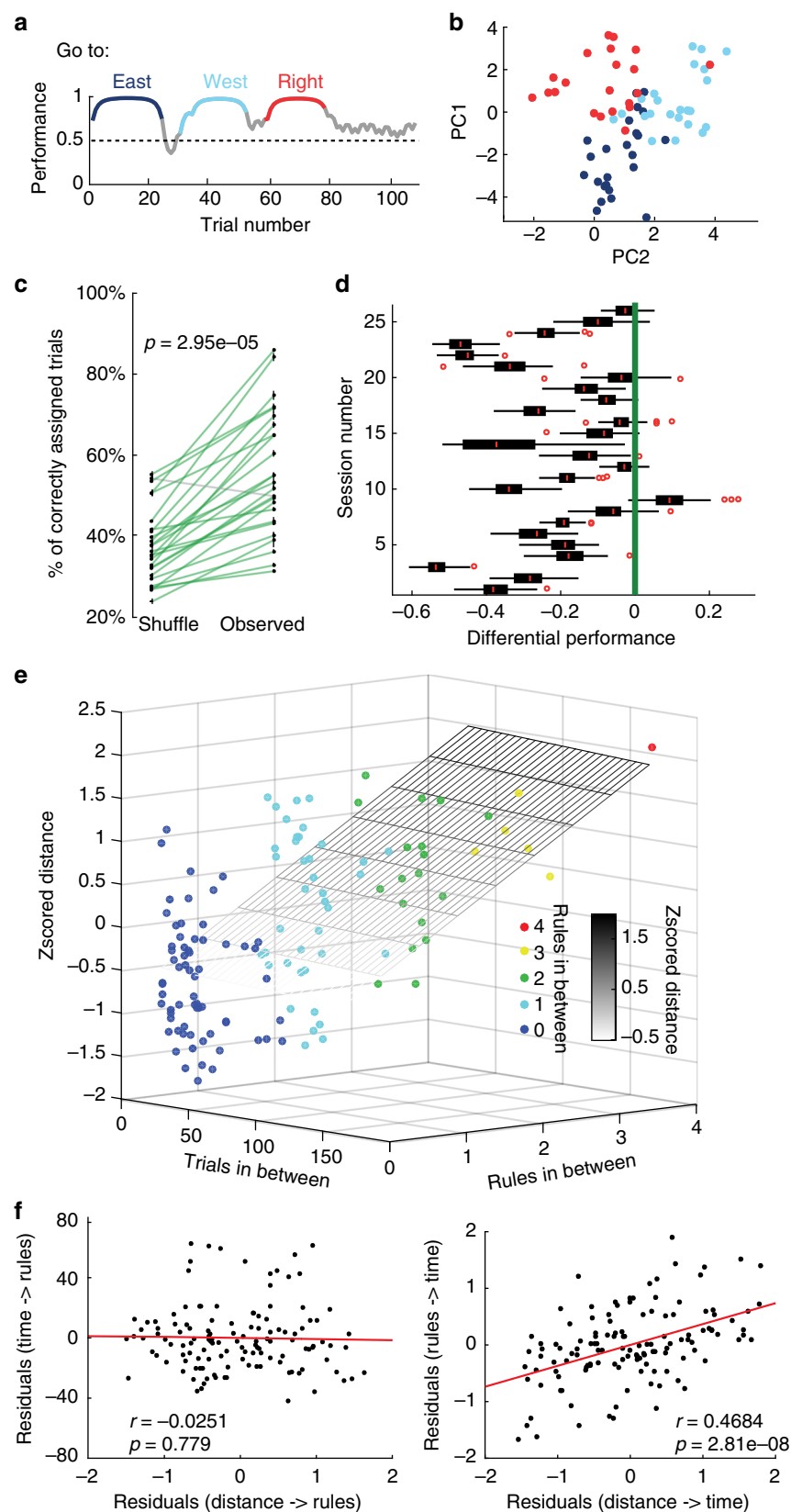

behaviourally required[2], denoting a flexible state. During the subsequent learning of the new rule, the representation of neuronal responses tends to be more stable and less sensitive to noise[53]. We show that firing rates of individual neurons are correlated with the behavioural performance of the rat during a rule switching task. We observed two groups of neuron with a different performance-related activity. For one group of neurons, the firing rate was negatively correlated with the performance of the animal, while for a second, complementary group of neuron, there was a positive correlation. Additionally, we also show that these correlations can be found without taking into account the firing information during reward, demonstrating that it is not only a reward-monitoring value of performance. These complementary neuronal assemblies with opposite correlations between firing rate and performance might reflect the simultaneous ability of the network to maintain rewarding behaviours and perform the task with high success or to induce flexible changes when the success rate is low.

We investigated neuronal population activity by representing each trial as a population vector constructed with the individual firing rates of simultaneously recorded neurons. We defined a specific population vector per trial and a "rule cluster" defined by the combination of points of all the trials belonging to a specific rule. Similar to Durstewitz et al[2], we show that vector populations of trials belonging to the same rule tended to be clustered together, in contrast to vector populations with an intrinsic random organisation. However, by examining the neuronal population dynamics using population vectors, we extended the rule-change analyses to multiple rule changes presented during the same recording session, including in some cases the repeated presentation of the same rule. Not only was each rule reflected by an independent state representation or "rule cluster" different from firing during other rules, but when the same rule was presented again later in that session, the second presentation fell into a new network state, different from the first presentation of this particular rule. Our results suggest that identical rules in a prefrontal cortex-dependent task will not lead to a similar neuronal encoding, but rather to a new set of prefrontal network activity. In line with Heraclitus's quote, we provide evidence that the same action plan is not "perceived" identically a second time, as the new experiences shaped a new cognitive state. One explanation for this phenomenon might be that firing rates of the prefrontal cortex are context dependent[11,39,54–57] and could be influenced not only by past events but also by multiple inputs that the prefrontal cortex is receiving from other areas[11,12,58]. This implies that even when the rule is the same in the second presentation, the actual context in which the rule has been presented differs from the previous presentation. Indeed, due to the fact that time has been shown to be responsible for state changes or drifts in the neuronal activity of the prefrontal cortex[39], we decided to test the contribution of time for the new encoding of the repeated rule. Interestingly, when corrected by the number of rules appearing,

time was no longer explanatory for the drift (Fig. 4f), but rather the number of multiple rules or rule changes. This leads to the conclusion that it is rather the experience of a new rule and the accompanied strategy switch that contribute to the appearing of new firing state rather than a monotonous drift of the neuronal activity over time. As a second possibility, reservoir neuronal networks related to prefrontal cortex models have been proposed and imply a non-stationary response of neurons during performance of a task[59,60], when neuronal activity is temporarily lifted from a random firing to a stable representation initiated by the requirements of the task. Such periods of modulation might depend on the capability of the network to decode information[61]. This would imply that the state into which the network is being driven depends on stochastic processes occurring when the network is not being recruited, bringing a different state representation to the same rule. However, time-dependent processes in the prefrontal cortex are not totally excluded by the reservoir networks, as it has been previously shown that a dynamic reservoir networks in the prefrontal cortex can maintain several distinct timescales of reward memory, thus incorporating previous information and facilitating flexible changes and adaptive reinforcement learning[62].

In conclusion, we show that individual neuronal firing rates correlate with the fluctuating performance of the rat during changes in strategy. The observed assemblies of those neurons portray the interaction of two groups of opposite but complementary patterns forming stable neuronal populations, which are being dynamically recruited for the purpose of flexible cognitive behaviour. Interestingly, when these populations formed different states for each rule presented, rules were not specifically encoded by a defined population vector as the same rule has two different representations when presented twice during the same session. Our analyses indicate that these observations are independent of the movement of the animals. This evidently advocates that neuronal firing in the prelimbic cortex reflects changes in strategy and task-performance monitoring but does not represent long-term strategies or rules permanently.

## Methods

**Experimental animals**. Five long Evans rats from Charles River Laboratories (male, 300–600 g), were kept in 12 h light cycle during behavioural experiments (performed during light cycle). All experimental procedures were performed under an approved licence of the Austrian Ministry of Science and the Medical University of Vienna.

**Surgery and microdrive implantation**. Animals were anaesthetised with Isoflurane (induction 4%, maintenance 1–2%; oxygen flow 2 l/min) and fixed on a stereotaxic frame, where body temperature was stabilised using a heating pad. Iodine solution was applied to disinfect the surgery site and eye cream was used to protect the corneas. Local anesthetic (xylocain® 2%) was used before the incision. In order to avoid dehydration, saline solution was injected subcutaneously every 2 h. Seven stainless steel screws were anchored into the skull to improve the stability of the construct and two of the screws were placed onto the cerebellum as references for the electrophysiological recordings. Subsequently, based on the rat brain atlas[63], a craniotomy was performed above the prefrontal cortex area (from

---

**Fig. 4** Firing rates of neurons change according to different rules followed by the animal. **a** During a behavioural session, the animal successfully adjusted its performance during three different rules presented consecutively. Highlighted episodes represent the trials used to generate the representations of network states. **b** Projection of multi-unit activity ($n = 24$ neurons) onto the two highest principal components for the session shown in (**a**). **c** Comparison between the performance of K-means clustering on the observed firing rates versus the performance of the k-means clustering on the shuffled firing rates ($p = 2.95e-5$, Wilcoxon signed-rank. **d** Normalised distributions of the performance (shuffle data minus observed data) for each of the 26 sessions, using the centre of mass as a starting point of the k-mean clustering algorithm. The green line references performance of the observed data. Red circles denote outliers. Box plots show median, 25th and 75th percentile. **e** Three-dimensional plot of a multiple linear regression with the z-scored Euclidean distance between clusters as the dependent variable. The regressors are the number of trials in between and the number of rules in between. The plane is the resulting equation of the regression. Note that the distances are better explained by the rules ($p = 2.81e-08$) than the number of trials ($p = 0.779$) in between. **f** Partial correlation plots of the distance between the centre of masses of the rules and the number of trials taking into account the number of rules (left) and the distance between centre of masses of the rules and the number of rules taking into account the number of trials (right)

bregma: + 3.25 mm anterior, 1 mm lateral, right hemisphere), where, after removal of the dura mater, an array of 12 independently movable, gold plated (100–500 kΩ) wire tetrodes (13 μm insulated tungsten wires, California Fine Wire, Grover Beach, CA) mounted in a microdrive (VersaDrive, State University of New York Downstate Medical Center) were implanted. Then, paraffin wax was applied around the tetrode array and the lower part of the microdrive was cemented (Refobacin® Bone Cement) to the scalp. At the end, the surgery site was sutured and systemic analgesia (metacam® 2 mg/ml, 0.5 ml/kg) was given. Animals were given post-operative analgesia (Dipidolor 60 mg diluted per 500 ml drinking water) and were allowed at least 7 days of recovery time.

**Maze description and behaviour**. A strategy switching task based on the consecutive learning of allocentric and egocentric spatial navigation rules[1] was used as a behavioural paradigm. The paradigm was modified to train the animal to perform multiple rule changes over the same session, by changing the rule unannounced

every time the criterion for rule learning was achieved (13 out of 15 correct choices). The maze consisted of a plus-maze (55 cm high, made of wood) composed of four arms (80 × 11 cm) with a 90° angle separation between them. Two opposite arms (named North and South) were starting arms and the other two (named East and West) were rewarded arms. The maze was located within four synthetic wall panels to maintain the uniformity of the environment. Three landmarks (triangle, circle, and square made of polystyrene) were visible and attached to different panels. Reward (sucrose pellets, 3 × 20 mg, TestDiet) was delivered by dispensers (Campden Instruments Ltd) . The entire automation of the maze (sensors and pellet feeder controls) was controlled by self-made scripts in MATLAB. Tracking of the rats' movement was monitored by triangulating the signal from three LEDs (red, blue, green) placed on the implanted headstage and recorded at 25 frames per second by an overhead video camera (Sony). One week after surgery, habituation to the maze started. Rats were food-restricted to maintain 85% of their weight. First, during two consecutive days, animals were placed inside

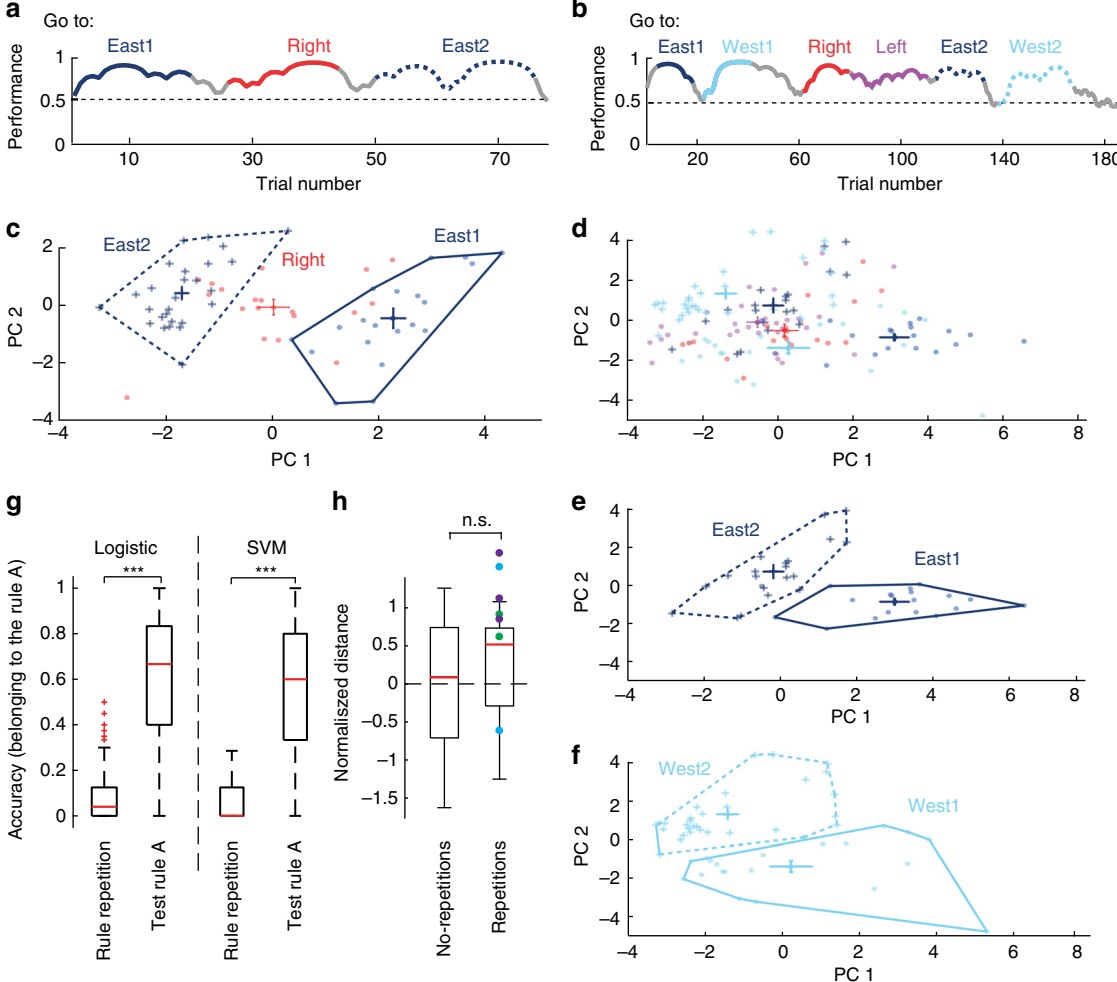

**Fig. 5** The repeated presentation of the same rule is not accompanied with a reemergence of the same prefrontal network state of firing. **a** Behavioural session during which the animal performed successfully during three consecutive rules; the first and the third rule applied were identical (go east). **b** Behavioural session with 7 rules presented. Two rules are presented twice at different times (go east and go west). Highlighted performance represents the trials used to calculate firing-states. **c** Projection of multi-unit activity (n = 12 neurons) onto the two highest principal components for the session shown in (**a**). **d** Projection of multi-unit activity (n = 18 neurons) onto the two highest principal components for the behavioural session shown in (**b**). Network states of different rules are colour coded and the centre of mass of each of them is presented with error bars. **e, f** Isolated and non-overlapping network states of two different rules (top and bottom), which were presented twice during the session example in b. Note that the rules presented twice at different times during the same session (dark and light blue, straight and dotted contours) resulted in different firing states of the network. **g** Comparison of the accuracy of two classifiers (logistic and support vector machine) trained to detect a given rule 'A'. Note that the classifiers can detect trials belonging the rule 'A', but failed at detecting its repetition 'A°' as part of rule 'A' (p < 1×10e−20 for both logistic and SVM classifiers, Wilcoxon rank-sum test). Box plots show median, 25th and 75th percentile. **h** Box plot indicating the normalised Euclidean distance measured between the centres of rules without dimensionality reduction. Red line indicates the median. Coloured dots indicate different animals (Green = HM02, Blue = HM06 and purple = HM07). Note that distances of rules with or without repetitions of the same rule were not significantly different (p = 0.58, Wilcoxon rank-sum test). Euclidean distances were normalised to account for a different number of rules which were presented between the same rule (see Methods). Box plots show median, 25th and 75th percentile

of the recording room for a period of around 1–3 h. From the third day on, rats were placed in a ceramic bin at the centre of the maze for an hour and were later positioned on the maze in order to start exploration. Some sugar pellets were placed on the maze and several more at the reward zone. Once the animals learned to explore the maze and consumed the sugar pellets (usually 1–2 days), they were habituated to the activation sound of the pellet feeders, by only giving reward once they entered the reward zone and activated the pellet feeder. After that, the training phase started. It consisted of manually placing the animals at one of the two possible starting points; in this case, the rats had to reach any of the two reward zones, where the reward was given indistinctly. After reward was consumed, rats were manually placed in the bin located at the centre of the maze. Once 30–50 trials were successfully performed by the rat, the full strategy switching task was presented on all subsequent sessions with either rewarded or non-rewarded trials, depending on the actual rule and decision. A typical recording day took place as follows: The rat was kept for 10 min inside of the ceramic bin at the centre of the maze, where baseline recordings were made. Then the task started and the initial

rule to be presented was given pseudo-randomly by the script used to control the maze and the task. The animal was taken from the bin and located at one of the two possible starting positions (North or South pseudo-randomly selected) and the bin, where the animal was before was placed to block the opposite starting arm. The rat then ran towards one of the two possible goal arms. After crossing a sensor located at the end of the arm, reward was given in case of correct. Independently of the correctness of the trial, the animal was taken 2 or 3 s later, placed again in the ceramic bin at the centre of the maze, and after 3–7 s the rat was placed again at one of the two possible starting arms and a new trial was started. The same starting arm could not be selected more than four consecutive times. During the task, when an animal achieved 13 out of 15 possible correct trials, the rule was changed without warning. Four different rules were presented in the task (Fig. 1a): two are allocentric rules (landmark-referenced) during which, independently of the starting point, animals have to go to a reward arm, and the other two rules are egocentric rules (self-referenced) during which, independently of the starting point, the animals should turn to their own right or left. A rule change within the same strategy

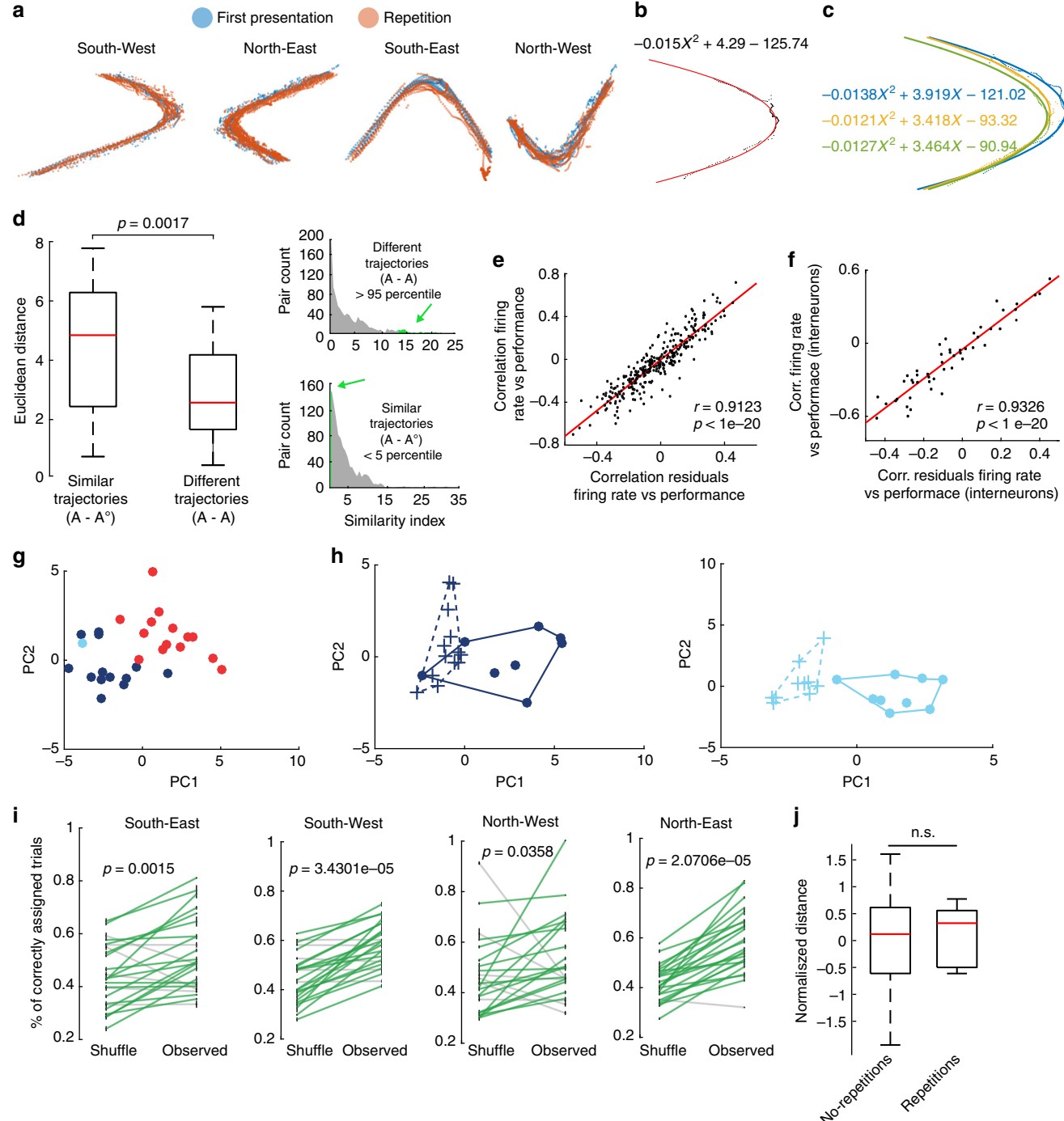

is defined as a reversal, while a rule change from one strategy to another is defined as a switch. The maze was cleaned with an odour-neutral solution every 10 trials to avoid odour-guided navigation. At least one behavioural switch (allocentric to egocentric or egocentric to allocentric rule learning) was successfully learnt during each session considered and further analysed. To define learning, a state-space model designed by Smith et al.[35] was used to analyse the outcome of the rat behaviour during the strategy switching task. Such a state-space model has been previously shown to be more reliable for performance prediction compared to other methods like the change-point test, moving average and the specified number of responses method[31]. The algorithm computes the probability and confidence intervals of the rat choosing the correct arm, taking into account the previous outcomes. The performance curve is defined as this probability. Thus, by using the lower confidence interval of the behavioural curve from the ongoing rule, three different behavioural phases were assigned to each trial (chance level is 0.5)[1]: naive (below 0.1 for reversals and 0.3 for switches); learning (between 0.1- and 0.6 for reversals and 0.3-0.6 for switches); and learnt (over 0.6).

The data presented here correspond to behavioural controls and electrophysiological recordings. Behavioural controls were performed in two animals with a total of 48 rules tested (12 allocentric with landmarks, 11 allocentric without landmarks, 12 egocentric with landmarks and 13 egocentric without landmarks) in 17 sessions performed. The electrophysiological data corresponds to 26 sessions performed by 3 animals (8, 12 and 6 sessions, respectively). In total, 66 rules changes were achieved (median number of changes per session = 3). For each session, the rules were selected pseudo-randomly (trying to secure at least a switch between strategies and a change of rules within the same strategy). The median number of trials per session is 98.5 trials.

**In vivo electrophysiology**. A headstage (HS-132A, 2 × 32 channels, Axona Ltd) was used to pre-amplify the extracellular electric signals from the tetrodes. Output signals were amplified 1000 × via a 64-channel amplifier and then digitised with a sampling rate of 24 kHz at 16-bit resolution, using a 64-channel analogue-to-digital converter computer card (Axona Ltd). Single-unit offline detection was performed by thresholding the digitally filtered signal (0.8 − 5 kHz) over 5 standard deviations from the root mean square on 0.2 ms sliding windows. For each single-unit, 32 data points (1.33 ms) were sampled. A principal component analysis was implemented to extract the first three components of spike waveforms of each tetrode channel[64].

Spike waveforms from individual neurons were detected using the KlustaKwik automatic clustering software[65]. Using the Klusters software[66], individual single units were isolated manually by verifying the waveform shape, waveform amplitude across tetrode's channels, temporal autocorrelation (to assess the refractory period of a single-unit) and cross-correlation (to assess a common refractory period across single-units). The stability of single-units was confirmed by examining spike features over time.

**Histology**. At the end of recordings, animals were anaesthetised with urethane and micro-lesions were made at the tip of the tetrodes by using a 30 μA unipolar current for 10 s (Stimulus Isolator, World Precision Instruments). Rats were perfused using saline solution, followed by 20 min fixation with 4% paraformaldehyde, 15% (v/v) saturated picric acid and 0.05% glutaraldehyde in 0.1 M phosphate buffer. Serial coronal sections were cut at 70 μm with a vibratome (Leica). Sections containing a lesion were Nissl-stained to verify the position of the tetrodes.

**Firing rates and correlations to performance**. Firing rates were determined in each recording session and for each neuron by dividing the number of spikes over the corresponding time of the trial. Every trial started the moment that the rat initiated movement from the starting arm towards the end arm. The trial can be divided into 3 different behavioural segments (run, reward, inter-trial), plus

another segment which consists on the entire duration of the trial. The run segment was defined as the period from the beginning of rat movement, at the starting arm, towards the reward arm, until the rat crossed a reward sensor located 10 cm at the terminal part of the rewarded area. The reward zone segment was defined as two seconds starting from the moment the animal crossed the reward sensor. The inter-trial segment was defined from the end of the reward segment until the beginning of the next run segment (it comprised a waiting period between 3 and 7 s inside of a bin, plus the time that takes to the animal to start moving and initiate the next trial). The entire trial spans from the start of a trial until the start of the next one. Therefore, in total, 4 firing rates were calculated by counting the spikes present during the specific segment and dividing it by the time spent in that segment. The average length (in seconds) for each segment (except Reward, which is always 2 seconds), specified as median, 25th and 75th percentile were: Run (2.86, 2.28, and 4.34), Inter-trial (19.16, 15.48, and 18.42) and the entire trial (24.3, 20.36, and 32.62).

Correlation of individual firing rates for each different segment and the entire trial were correlated to the observed performance of the animal using Spearman correlation. In addition, Bonferroni–Holm correction was used to correct for false positives. For the comparison made in Fig. 2e, shuffled data were generated. If given a matrix of firing rates $FR_{mxn}$, where '$m$' are the trials and '$n$' the number of neurons, together with a response vector ('1' is correct, '0' is incorrect) $RV_{mx1}$, shuffle was performed by rearranging the rows of both the firing rate matrix and the response vector, therefore disarranging the temporal organisation of the recording session but keeping the relation between firing rate and responses. The performance was recalculated over this new shuffled data, generating a shuffled performance curve, which was later correlated with the firing rates.

**Neuronal state-space**. Neuronal state-space was formed by defining each trial as a function of the $z$-scored firing rates of the neurons recorded during that session $T_m = FR_{1m}, FR_{2m}, ..., FR_{nm}$, where $FR_{nm}$ is the firing rate of neuron '$n$' in that specific trial ($T_m$). The vector of neuronal activity for a given trial was often referred in the text as a population vector. Only trials from the learning and learnt periods were taking into account.

**Euclidean and Mahalanobis distance**. Calculation of distance between two points was a recurrent analysis in the manuscript. Both Euclidean (1) and Mahalanobis (2) distances were used, ending up with similar results. Euclidean distance is defined as follows:

Given two trials (or centre of masses of rule clusters) $T_a$ and $T_b$, defined by the firing rate (FR) of all neurons '$n$' at that trial $T_a = (FR_{1a}, FR_{2a}, ... , FR_{na})$ and $T_b = (FR_{1b}, FR_{2b}, ... , FR_{nb})$ the Euclidean distance between the two trials is:

$$ED(T_a, T_b) = \sqrt{\sum_{i=1}^{n}(T_{ia} - T_{ib})^2} \qquad (1)$$

Mahalanobis distance was used mainly between a trial and a rule cluster; therefore, it is defined as follows:

Given a trial $T_a = (FR_{1a}, FR_{2a}, ... , FR_{na})$ and a rule cluster with mean $RM = (MFR_1, MFR_2, ... , MFR_n)$ where $MFR_n$ is the mean firing rate of a cell ($n$) over all the trials conforming the rule cluster, the Mahalanobis distance between the trial and the rule cluster is:

$$MD(T_a, RM) = \sqrt{(T_a - RM)' S^{-1}(T_a - RM)}, \qquad (2)$$

**Fig. 6** Trajectory and speed of the animal do not explain the different neuronal network representation of rules and its repetitions. **a** Overlapping running-trajectories during a rule and during the later repetition of the same rule. **b** Trajectory during one trial (black) and the fitted parabola (red). **c** Different trajectories in the same recording day and their fittings. **d** Right, similarity index distributions calculated for all possible pairs between trajectories during trials within a rule 'A' (top) and the possible pairs between trajectories during a rule 'A' and its repetition 'A°' (bottom). Green denotes similar (<5 percentile) and different (>95 percentile) pairs. Left, comparison of the Euclidean distance in the network firing state for trials with similar trajectories between a rule 'A' and its repetition 'A°' and for trials with different trajectories belonging only to rule 'A' ($p = 0.0017$, Wilcoxon rank-sum test). Box plots show median, 25th and 75th percentile. **e** After removing the effect of the trajectory and the speed of the animal, residuals remain correlated to performance, similar to neuronal firing rates ($n = 300$, $p < 1e−20$, $r = 0.9123$, Spearman correlation). **f** Same as (**e**) but for firing rates of putative interneurons ($n = 50$, $p < 1e−20$, Spearman correlation). **g** Projection of the multi-unit activity residuals onto the first two principal components for the session shown in Fig. 5a. **h** Isolated and non-overlapping network states (using the residuals) during two different rules (left and right) which were presented twice during the session shown in Fig. 5b. **i** Comparison between the performance of k-means clustering on firing rate residuals vs. the performance of the k-means clustering on the shuffled firing rate residuals, for the four possible trajectories (South-East, $p = 0.0015$; South-West, $p = 3.43e−05$; North-West, $p = 0.0358$; North-East, $p = 2.07e−05$. Wilcoxon signed-rank test). **j** Box plot indicating the normalised Euclidean distance measured between the centres of rules without dimensionality reduction in the network state formed by the firing rate residuals. Red line indicates the median. Distances of rules with or without repetitions of the same rule were still not significantly different ($p = 0.40$, Wilcoxon rank-sum test). Box plots show median, 25th and 75th percentile

where the operator (') is the transpose and $S^{-1}$ is the inverse of the covariance matrix.

**Similarity index**. The similarity index was defined as the normalised Euclidean distance between the coefficients of the quadratic equation modelling a trajectory. The higher the index, the more different are the trajectories.

Any given trajectory (TR) in a trial ($m$) can be represented with the coefficients of a quadratic equation:

$$TR_m = a_m x + b_m x + c_m \tag{3}$$

The normalised Euclidean distance (nED) between two trajectories is defined as:

$$nED(TR_1, TR_2) \sqrt{W_a(a_2 - a_1)^2 + W_b(b_2 - b_1)^2 + W_c(c_2 - c_1)^2}, \tag{4}$$

where $W_a = \frac{1}{\max(a_m) - \min(a_m)}$, $W_b = \frac{1}{\max(b_m) - \min(b_m)}$, $W_c = \frac{1}{\max(c_m) - \min(c_m)}$

**Clustering**. Clustering was performed using the K-means clustering algorithm. This method requires the number of clusters to be given, and initialisation of the cluster centres (one per cluster to be found). Then, we computed the distance from each point to each centroid and assigned the point to the centroid from which the shortest distance is measured. Thus, a new centroid is calculated from the newly assigned points and the iteration continued until it found a balance (no change in the assignment of new points). For the data in Fig. 4c, we used unsupervised clustering based on K-means. We assigned each cluster one rule according to the majority of times this cluster represented a state corresponding to that rule. We compared the assigned rule values to the actual rule values and created an accuracy percentage by dividing the positively assigned rule values by the total number of points. We calculated the accuracy for both the observed and shuffled data.

**Classifiers**. A logistic regression and a support vector machine with linear kernel were used to create two classifiers which classify a trial (represented by the $z$-scored firing rate of all the neurons during that trial) to belonging to a given rule 'A'. The following procedure was done in each recording session, which contained a rule repetition ('A' as the rule and 'A°' as the rule repetition): Classifiers were trained with a set consisting of random selection of 70% of the trials of rule A, labelled as '1', and the rest of trials of the other rules labelled as '0'. Trials of the rule repetition 'A°' were not included in the training set. Classifiers were later tested by a data composed of the trials in the rule repetition 'A°' and the remaining 30% of trials of rule 'A', not used in the training set. Due to the random assignment of trials of rule 'A', to both training and test set, the procedure was repeated 1000 times. Figure 5g shows the classification of the test data as being part of rule A, divided by either the tested trials in 'A' or the trials in the repetition 'A°' (Accuracy of belonging to rule 'A').

All calculations were made in MATLAB (Mathworks, version R2009b and R2015b.) and statistical analyses were performed with MATLAB and Microsoft Excel. All the statistical tests used were non-parametric with Bonferroni–Holm correction unless stated differently in the text. The data used in linear regressions were checked for homoscedasticity. The principal component reduction was performed using a dimensionality reduction toolbox for Matlab[67].

**Code availability**. The computer code that supports the findings of this study is available from the corresponding authors upon reasonable request.

**Data availability**. The data that support the findings on this study are available from the corresponding authors upon reasonable request.

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

### Acknowledgements

This work was supported by the Vienna Science and Technology Fund (WWTF), project LS14-095. We thank E.Borok and R.Hauer for excellent technical support; F.Stella, B.Lasztoczi and T.Oezdemir for commenting on a previous version of the manuscript and M.Lagler for comments on the analysis.

### Author contributions

H.M.-V., S.C., J.P., G.D. and T.K. contributed to experiments, data analysis, and pre-paration of the manuscript.

### Additional information

**Competing interests:** The authors declare no competing financial interests.

