## [Peer Review File · Nature Communications]

Reviewers' comments:

Reviewer #1 (Remarks to the Author):

Fluid network dynamics in the prefrontal cortex during multiple strategy switching

H. Malagon-Vina, S. Cioocchi, Johannes Passecker, Georg Dorffner, T. Klausberger

Summary

The authors trained 5 rats (2 for behavior, 3 for ephys) on a fairly difficult rule-switching task. The task involved both within-strategy shifts (e.g. go east to go west) and across strategy shifts (e.g. go west to go right). Rats perform a block of a single rule until they achieve good performance (13/15 trials correct) and then an unsigned switch occurs. They recorded using 12 tetrodes to examine PFC activity in the task. There are several claims made in the paper: (1) PFC neurons track the animals performance, (2) Activity is relatively stable (or clustered) inside each block, and shifts across blocks (i.e. when the rules change) (3) That when animals return to an earlier rule in a single session, the population activity does not return to its earlier configuration, but enters a new state, no different than if it was a rule unexperienced on that day.

The general questions of the paper are interesting and the paper is clearly written but the most novel claim of the paper (#3 above) is not supported by the current analyses, therefore I cannot recommend it for publication in its current form.

Major Issues

Low N / No reports of reliability of findings across animals.

The authors do a good job at showing the population but they never show which data came from which animal. With only a small # of sessions per animal it will be hard to see significant effects for individual animals, but it would be good to show. Eg. in 5c each animal could be a different color line and in 5d the sessions could be sorted according to animal and then time.

No behavioral control

This is really the biggest issue with the paper. It is well known that PFC activity is highly correlated with movements¹. The authors make no attempt to determine how much variability in their observed neural activity is accounted for by the trajectory of the animal. The authors cluster all trials (both correct and error) in a session. It looks like, from the examples, that performance gets worse over the session (need to show this data). If that is the case, then the 2nd "east" block is not the same as the 1st "east" block. Even if performance (measured as % correct) was the same, the trajectory of the animals might change. Authors need to show the trajectories, including run speed, etc. as in 1. After taking account all variance that is due to actual movements then the authors could try to ask if the residuals in activity has consistent or inconsistent representation of strategy.

But still, the PCA analysis and the euclidean distance analyses are unconvincing. Why? Imagine that PFC keeps track of the rule but also the # of blocks performed. This extra block # information would lead the first "east" block of activity to cluster away from the 2nd "east" block of activity.

What would be more convincing is to train a decoder (e.g. SVM or even a logistic regression) on the first two rule blocks (e.g. as "east" and "not east" labels) and then test them on later rule blocks. If the decoder succeeds, that is evidence for preserved representation. if the logistic decoder succeeds then the rule representation is linearly encoded. With SVM you could find a more complex

representation of rule. If it fails, that is evidence against preserved representation. However, failure to decode could also be the result of low # of simultaneous units, firing rate, etc. Some simulations would be appropriate.

There are other small issues with the paper, but a manuscript re-written to deal with the behavioral control issue will be different enough that it isn't of much value to discuss the other things.

Euston, David R., and Bruce L. McNaughton. "Apparent Encoding of Sequential Context in Rat Medial Prefrontal Cortex Is Accounted for by Behavioral Variability." *Journal of Neuroscience* 26, no. 51 (December 20, 2006): 13143–55. doi:10.1523/JNEUROSCI.3803-06.2006.

Reviewer #2 (Remarks to the Author):

Malagon-Vina and colleagues recorded for rat medial PFC neurons during performance of a multiple rule switching task. They isolated out neurons whose firing rates varied with behavioral performance. They then examined how well these neurons formed representations of different rules and found that as ensembles they were better than as individual neurons. They also found that when the same rule was encountered later in the session, the ensemble representation moved to a novel state indicating a fluid encoding of rules dependent upon experiential/contextual factors. This dataset is very strong and the main findings of this paper will make for an important contribution to the literature. This depiction of mPFC ensembles as encoding a fluid/flexible yet organized representation of the cognitive palette fits well into a growing body of literature that is helping to explain the mPFC's role in behavioral and cognitive flexibility. However, I do have some concerns that need to be addressed that will help readers better follow the paper and it's myriad analytical steps.

Most of my major concerns are related to the presentation of the analyses. Overall, the authors need to do a much better job 'walking' the reader through all these layered analyses presented here. A more direct narrative writing style may help to explain why certain analyses were chosen. Also, the behavioral analysis is not thorough enough for the claims the authors want to make later in the paper (for specific comments see below).

Lack of important details about specific analyses (detailed below).

While I acknowledge that this will be tough, but the authors need to find a way to show better how the raw data relates to their findings. Maybe some single unit examples that are a bit more intuitive than found in Figure 2. Raster plots or PSTH's which show firing over the trial itself and over the sequence of trials/rules.

Specific concerns:

Results – Please add a brief, more casual walk through of the task either at the top of this section or at the end of the Intro to help the reader. The description in Fig1 is not entirely clear and it's not clear to this reader what a typical recording session was like in terms of the sequence of goal locations/strategies. I think an example session detailing what rules were in place for what trial numbers would definitely help.

The delay period between trials is not clear to me. The methods say the animal was placed in a bin for 3-7 seconds. Was this bin on the maze? Was it clear? Was the bin in the same place on every trial? Or was it just a trap door sequestering the animal in the last reward arm? This needs to be much clearer.

Also, a diagram showing how the task is broken down into behavioral segments would help better follow the ephys analyses.

More behavioral analysis is needed to better understand what neuron firing rates correlating with performance means. Does running speed vary similarly? Is this true for correct and error 'reward arms'? While I see the importance of the early cell selection method, I think there are too many other behavioral variables that could account for these changes besides % correct. Honestly, I don't think that would change my interpretation of the data, but it would still be nice to see the performance metric more justified as the ultimate unit to behavior analyses.

I don't understand how trials were shuffled by trial number and the associated corr/inc and FR were kept but the correlation values changed. If you just took the same data and reordered all variables together, why does the corr value change? If trial 30 has a FR of 5Hz and a performance prob of 0.8, it shouldn't matter if trial 30 is changed to being trial 1. This is what it sounds like the authors did here from their description. That clearly can't be true and I don't believe this was at all the authors intentions. Nonetheless, I'm still not certain what was shuffled. Presumably the FR and performance probs were shuffled independently to show the relationship was not random. That doesn't address the idea about "instantaneously-received reward or lack thereof" or other possible behavioral factors that could be driving FR changes besides accumulated reward history.

Fig 3 results could easily be due to behavioral differences between error and correct trials. A more detailed description of the behavior is needed here as well. Which behavioral segment is this analysis performed on? On correct trials animals are presumably stopping their movements to retrieve and consume the reward, while on error trials they are likely searching for reward or just merely moving around more. Generally, high FR cells in the PFC have firing rates highly correlated to movement, so this finding could be indicative of a behavioral and not cognitive change.

"Neuronal firing rates are predictive of task-performance" Why not compare the actual data correlations to the permuted data correlations, thus creating a unique empirical distribution for each ensemble? This would be more prudent analysis given the unknown nature of the distribution of predicted vs actual outcomes.

L 144 – The statement "This demonstrates that the relation between firing rate and performance is better encoded by a population of cells" is not really backed up by more than an apples and oranges comparison. This is an interesting point and indeed recent publications have shown striking differences between single unit and ensemble information coding (Hyman et al., 2012; Ma et al., 2015; Hyman et al., 2017), but that is not really what was being compared here. An LM prediction of performance compared with actual performance is not the same as FR compared with performance. More analysis of ensemble/single unit differences is needed to support this strong a claim. That or make the claim more suggestive in nature and cite past articles showing this effect.

L149 - typo "remains"

"Neuronal firing patterns in the prefrontal cortex reflect changes in rules and strategies" – what time bins were used for this analysis? Did the authors just take a single mean for the entire trial? Each behavioral segment? A series of time bins for each trial for each cell? Without this basic information it is impossible to accurately interpret the state space.

L 217 – use of the word "determined" in this sentence is awkward.

The authors consistently use Euclidean distance measures, when Mahalanobis distance (which also

incorporates the covariance matrix) is usually used in these analyses. Why did the authors make this choice? Especially, in light of the acknowledged modeling of the analytical approach on the Durstewitz et al., 2008.

First paragraph of the Discussion – the authors refer to “two complementary neuronal groups” – presumably these are the neurons that are either positively or negatively correlated with behavior. These groups are not mentioned in other analyses. Were the ensembles split for the state space analyses? Is this just sloppy writing or is something not clear in the Methods?

The discussion of reservoir network models is nice at the end of the Discussion section, but it fails to bring up temporally driven reservoir models like those supported by primate recordings (Bernacchia et al., 2013). Considering the authors bring these models up as a counter to time-driven network changes, this might allow for a nice discussion of how to incorporate both ideas.

The “Analyses and Statistics” section of the Methods is poorly organized, choppy, and lacking. For involved layered analyses like this it generally best to have the Methods section read more narratively, while obviously still containing all the necessary details. This way, what analysis choices were made and why can be justified

We thank the referees for their overall positive evaluation of our manuscript and for their most helpful suggestions and questions. We have performed additional analysis and improved the manuscript according to their comments.

Point-to-point reply to referees:

Reviewer #1 (Remarks to the Author):

Fluid network dynamics in the prefrontal cortex during multiple strategy switching

H. Malagon-Vina, S. Ciocchi, Johannes Passecker, Georg Dorffner, T. Klausberger

Summary

The authors trained 5 rats (2 for behavior, 3 for ephys) on a fairly difficult rule-switching task. The task involved both within-strategy shifts (e.g. go east to go west) and across strategy shifts (e.g. go west to go right). Rats perform a block of a single rule until they achieve good performance (13/15 trials correct) and then an unsignaled switch occurs. They recorded using 12 tetrodes to examine PFC activity in the task. There are several claims made in the paper: (1) PFC neurons track the animals performance, (2) Activity is relatively stable (or clustered) inside each block, and shifts across blocks (i.e. when the rules change) (3) That when animals return to an earlier rule in a single session, the population activity does not return to it's earlier configuration, but enters a new state, no different than if it was a rule unexperienced on that day.

The general questions of the paper are interesting and the paper is clearly written but the most novel claim of the paper (#3 above) is not supported by the current analyses, therefore I cannot recommend it for publication in its current form.

Major Issues

Low N / No reports of reliability of findings across animals.

The authors do a good job at showing the population but they never show which data came from which animal. With only a small # of sessions per animal it will be hard to see significant effects for individual animals, but it would be good to show. Eg. in 5c each animal could be a difference color line and in 5d the sessions could be sorted according to animal and then time.

We acknowledge the concerns of the referee and have now have included individual animal data in the main figures and in the Supplementary figures. This indicates that the data does not change depending on the animal. The newly added data involves the following plots and tables:

- Supplementary Fig. 2 shows the number of significantly correlated neurons to performance per animal (HM02, HM06 and HM07) in all possible segments (Entire, Reward, Inter-Trial, Run). This corresponds to the data in figure 2b, of the main manuscript, segregated by animal.
- Supplementary Fig. 3a shows the comparison between the performance of the k-means clustering on the observed data vs the shuffled data separated by animals. In addition, Supplementary Figure 3b shows the normalized distributions of the

performance (observed vs shuffled) colour coded by animal. These plots correspond to the ones found in figure 4c,d on the main text.

- Supplementary table 1 shows the p-values (per animal) of the general linear model where the dependant variable is the distances between clusters and the regressors are the number of clusters and the number of trials in between. This table corresponds to the data in figure 4e of the main manuscript.
- We added in figure 5h the points for each animal in the boxplot for rule repetitions.

Reviewer # 1: No behavioral control.

This is really the biggest issue with the paper. It is well known that PFC activity is highly correlated with movements¹. The authors make no attempt to determine how much variability in their observed neural activity is accounted for by the trajectory of the animal. The authors cluster all trials (both correct and error) in a session. It looks like, from the examples, that performance gets worse over the session (need to show this data). If that is the case, then the 2nd “east” block is not the same as the 1st “east” block. Even if performance (measured as % correct) was the same, the trajectory of the animals might change. Authors need to show the trajectories, including run speed, etc. as in 1. After taking account all variance that is due to actual movements then the authors could try to ask if the residuals in activity has consistent or inconsistent representation of strategy. But still, the PCA analysis and the euclidean distance analyses are unconvincing.

We thank the reviewer for this insight which brings stronger and more convincing results when approached.

We have performed a series of different analysis that we have now included as a new chapter at the end of the results section (**‘Differences in speed and trajectory of the animal do not explain different firing states during different rules’**). Here we gave a summary of these analyses.

First, in Supplementary Fig. 5, we are showing an overlap of all the trajectories from the rule repetitions. These examples bring a visual check of the similarity of the trajectories between the trials with the initial rule ‘A’ and its repetition ‘A’ later in the session. Visually, hardly any difference between trajectories can be seen. However, further controls were made.

Taking advantage of the maze’s geometry, we fitted the trajectories to a quadratic equation:
 $Trajectory_{Trial} = ax^2 + bx + c$

This three coefficients can describe the trajectory of an animal in a trial. Figure 6a shows an example of the fitting. The fit also quantifies differences in trajectories (Figure 6b).

We compared the distances of trials which have very similar trajectories (using the three coefficients) between the first presentation of a rule ‘A’ and its repetition ‘A’ vs the distances of trials which have very different trajectories all within the rule ‘A’ (Similarity index was calculated as the normalised Euclidean distance between the 3 coefficients of the fitting. The

closer to 0, the more similar two trajectories are). Figure 6c shows that even though the trajectories between 'A' and 'A°' are very similar, the distances are significantly larger ($p = 0.0017$, Wilcoxon rank-sum test) than those between points in 'A' from very different trajectories. This implies that the trajectories *per se* do not explain the separated clustering of the rule repetition.

However, as the referee has suggested, trajectories and speed still might have a more subtle effect in the data. For this, we follow the suggestion of the referee to work with residuals and we modeled the firing rate of a neuron by using the coefficients of the quadratic equation and the speed of the animal in the following way:

$FR_T = \beta_0 + a_T\beta_1 + b_T\beta_2 + c_T\beta_2 + s_T\beta_3$ where FR is the firing rate; a,b,c are the coefficients of the quadratic equation and 's' is the speed in m/s, in a given trial 'T'. We found close to 30% of the neurons significantly correlated to at least one of the coefficients in the equation, which validates the approach. Then, we re-did all the analyses of the manuscript, but using the residuals of the firing rates. The new added result section shows that the observations made still hold true (Figure 6). All the analyses done with the residuals have similar results to our previous observations. Given that residuals are the part of the firing rate that is not explained by the trajectory or the speed of the animal, we confirm that the effects seen in the data are not a unique product of the animals movement.

Reviewer # 1 (continuation) Why? Imagine that PFC keeps track of the rule but also the # of blocks performed. This extra block # information would lead the first "east" block of activity to cluster away from the 2nd "east" block of activity.

What would be more convincing is to train a decoder (e.g. SVM or even a logistic regression) on the first two rule blocks (e.g. as "east" and "not east" labels) and then test them on later rule blocks. If the decoder succeeds, that is evidence for preserved representation. If the logistic decoder succeeds then the rule representation is linearly encoded. With SVM you could find a more complex representation of rule. If it fails, that is evidence against preserved representation. However, failure to decode could also be the result of low # of simultaneous units, firing rate, etc. Some simulations would be appropriate.

We thank the reviewer for the excellent idea. We have incorporated the results of the decoders in Figure 5g. The added results to the main manuscript read as follow:

To further corroborate that the neuronal population response of the repetition 'A°' of a given rule 'A' is distinct, we trained two different classifiers: a logistic regression and a support vector machine. The data was divided into a training set and a test set. The training set consisted of 70% of the trials corresponding to the rule 'A' and it was labeled as '1', and the trials of all other rules (except the repetition 'A°') which were labeled as '0'. The test set included the repetition 'A°' and the remaining 30% of the trials in 'A'. After building the decoder with the training set, an accuracy value of belonging to the rule 'A' is computed for the test set (number of trials classified as '1' over the total number of trials). The classifiers correctly assigned the data belonging to rule 'A' as '1' (belonging to rule 'A'), while the data of rule 'A°' is assigned as '0', indicating a different rule form 'A' (Fig. 5g, SVM -> $p < 1e-20$,

Logistic Regression $\rightarrow p < 1e-20$). These analyses suggest that the neuronal firing state during a rule repetition is different from the firing during the initial rule.

In general, we hope that with the new analyses that we have presented, we have shown that even though there is movement information in the firing rates of the prelimbic neurons, this information is not affecting our claims.

Reviewer #2 (Remarks to the Author):

Malagon-Vina and colleagues recorded for rat medial PFC neurons during performance of a multiple rule switching task. They isolated out neurons whose firing rates varied with behavioral performance. They then examined how well these neurons formed representations of different rules and found that as ensembles they were better than as individual neurons. They also found that when the same rule was encountered later in the session, the ensemble representation moved to a novel state indicating a fluid encoding of rules dependent upon experiential/contextual factors. This dataset is very strong and the main findings of this paper will make for an important contribution to the literature. This depiction of mPFC ensembles as encoding a fluid/flexible yet organized representation of the cognitive palette fits well into a growing body of literature that is helping to explain the mPFC's role in behavioral and cognitive flexibility. However, I do have some concerns that need to be addressed that will help readers better follow the paper and it's myriad analytical steps.

Most of my major concerns are related to the presentation of the analyses. Overall, the authors need to do a much better job 'walking' the reader through all these layered analyses presented here. A more direct narrative writing style may help to explain why certain analyses were chosen. Also, the behavioral analysis is not thorough enough for the claims the authors want to make later in the paper (for specific comments see below).

Lack of important details about specific analyses (detailed below).

While I acknowledge that this will be tough, but the authors need to find a way to show better how the raw data relates to their findings. Maybe some single unit examples that are a bit more intuitive that found in Figure 2. Raster plots or PSTH's which show firing over the trial itself and over the sequence of trials/rules

We consider that the suggestion of the reviewer is on point and some example raster plots will help to get a better feeling of the data. We have included the raster plots of examples neurons, showing the modulation of the performance. We decided to do rasters plots centered around the reward, extending 2 seconds to each side. (Supplementary Fig. 1)

Reviewer #2: Results – Please add a brief, more casual walk through of the task either at the top of this section or at the end of the Intro to help the reader. The description in Fig1 is not entirely clear and it's not clear to this reader what a typical recording session was like in terms

of the sequence of goal locations/strategies. I think a example session detailing what rules were in place for what trial numbers would definitely help.

The delay period between trials is not clear to me. The methods say the animal was placed in a bin for 3-7 seconds. Was this bin on the maze? Was it clear? Was the bin in the same place on every trial? Or was it just a trap door sequestering the animal in the last reward arm? This needs to be much clearer. Also, a diagram showing how the task is broken down into behavioral segments would help better follow the ephys analyses.

We apologise for the lack of clarity in our descriptions. We have improved the description of the task and the entire behavioural session in the results and throughout the manuscript, plus a more detail description in the methods section which answers your questions. In summary:

- **Was this bin on the maze?, Was the bin in the same place on every trial? Or was it just a trap door sequestering the animal in the last reward arm?** : The bin was in the centre of the maze during that period. When the trial started, the same bin was used to block the opposite starting arm, leaving only the goal arms as options.

- **Was it clear?**: It was not. It was a ceramic bin.

Reviewer #2: More behavioral analysis is needed to better understand what neuron firing rates correlating with performance means. Does running speed vary similarly? Is this true for correct and error 'reward arms'? While I see the importance of the early cell selection method?, I think there are too many other behavioral variables that could account for these changes besides % correct. Honestly, I don't think that would change my interpretation of the data, but it would still be nice to see the performance metric more justified as the ultimate unit to behavior analyses.

We understand the concerns of the reviewer, which fall in line with the concerns of reviewer #1. To address them, we have now included as a new chapter at the end of the results section (**'Differences in speed and trajectory of the animal do not explain different firing states during different rules'**). Here we gave a summary of these analyses:

First, in Supplementary Fig. 5, we are showing an overlap of all the trajectories from the rule repetitions. These examples bring a visual check of the similarity of the trajectories between the trials with the initial rule 'A' and its repetition 'A^o' later in the session. Visually, hardly any difference between trajectories can be seen. However, further controls were made.

Taking advantage of the maze's geometry, we fitted the trajectories to a quadratic equation:
$$\text{Trajectory}_{\text{Trial}} = ax^2 + bx + c$$

This three coefficients can describe the trajectory of an animal in a trial. Figure 6a shows an example of the fitting. The fit also quantifies differences in trajectories (Figure 6b).

We compared the distances of trials which have very similar trajectories (using the three coefficients) between the first presentation of a rule 'A' and its repetition 'A'' vs the distances of trials which have very different trajectories all within the rule 'A' (Similarity index was calculated as the normalised Euclidean distance between the 3 coefficients of the fitting. The closer to 0, the more similar two trajectories are). Figure 6c shows that even though the trajectories between 'A' and 'A'' are very similar, the distances are significantly larger ($p = 0.0017$, Wilcoxon rank-sum test) than those between points in 'A' from very different trajectories. This implies that the trajectories *per se* do not explain the separated clustering of the rule repetition.

However, as the referee has suggested, trajectories and speed still might have a more subtle effect in the data. For this, we follow the suggestion of the referee to work with residuals and we modeled the firing rate of a neuron by using the coefficients of the quadratic equation and the speed of the animal in the following way:

$FR_T = \beta_0 + a_T\beta_1 + b_T\beta_2 + c_T\beta_2 + s_T\beta_3$ where FR is the firing rate; a,b,c are the coefficients of the quadratic equation and 's' is the speed in m/s, in a given trial 'T'. We found close to 30% of the neurons significantly correlated to at least one of the coefficients in the equation, which validates the approach. Then, we re-did all the analyses of the manuscript, but using the residuals of the firing rates. The new added result section shows that the observations made still hold true (Figure 6). All the analyses done with the residuals have similar results to our precious observations. Being these residuals the part of the firing rate that is not explained by the trajectory or the speed of the animal, we confirm that the effects seen in the data are not a unique product of the animals movement.

In addition, even though the speed of the animal is also taking into account in the model and analyses previously described, we would like to let you know that 10 out of 26 sessions have a significant correlation between speed and performance. However, when we calculated the partial correlation for each neuron firing rate to the performance while controlling by the effect of the speed, the tendency of the correlations was not affected (see the next figure). In total 91 neurons were still correlated to performance, from which 71 were correlated previously without controlling speed. The following figure shows that the correlation values between the performance and the individual neuron firing rates is not different from the correlation value between the performance and the individual neuron firing rates controlled by the speed of the animal ($r = 0.9661$, $p < 1e-20$, Spearman correlation), suggesting that even though the speed affects the firing rate, it is not the only cause of the neuronal firing rate correlation to performance.

Overall we acknowledge the point of reviewer #2 in asking for a better consideration of the movement and other behavioural aspects of the behaviour to account for the correlations and we think these analyses clarify and bring the understanding that even though there is certain contribution of these variables to the firing rate, the correlation of the individual neurons firing rate to performance is still present once the other aspects are taking into account.

Reviewer #2: I don't understand how trials were shuffled by trial number and the associated corr/inc and FR were kept but the correlation values changed. If you just took the same data and reordered all variables together, why does the corr value change? If trial 30 has a FR of 5Hz and a performance prob of 0.8, it shouldn't matter if trial 30 is changed to being trial 1. This is what it sounds like the authors did here from their description. That clearly can't be true and I don't believe this was at all the authors intentions. Nonetheless, I'm still not certain what was shuffled. Presumably the FR and performance probs were shuffled independently to show the relationship was not random. That doesn't address the idea about "instantaneously-received reward or lack thereof" or other possible behavioral factors that could be driving FR changes besides accumulated reward history.

We apologize for the confusion, and we have improved the explanation in the main manuscript. In summary, once we shuffled the trials, the performance probability curves were calculated again. Let's say that trial 30 (which it was a correct one) has an associated firing rate of 5Hz, and an instantaneous performance of 0.8. When the trials are shuffled, this trial 30 will be now trial 12, still it will be correct and it will still have a firing rate of 5Hz, however, the performance will depend on the newly calculated curve, because the temporal organisation of correct/incorrect has changed. This implies that in case that the firing rates are only following correct and incorrect trials, the new correlation of the permuted data will not be very different from the observed ones.

Reviewer #2: Fig 3 results could easily be due to behavioral differences between error and correct trials. A more detailed description of the behavior is

needed here as well. Which behavioral segment is this analysis performed on? On correct trials animals are presumably stopping their movements to retrieve and consume the reward, while on error trials they are likely searching for reward or just merely moving around more. Generally, high FR cells in the PFC have firing rates highly correlated to movement, so this finding could be indicative of a behavioral and not cognitive change.

The analyses are performed on the entire firing rate, meaning that we include the run, reward and inter-trial segment. We understand the concerns of the reviewer and we have performed extra-analyses to be sure that neither the movement nor the behaviour during reward are affecting our claims. First, we use the firing rate residuals (after modeling for movement and speed as previously described) and repeat the correlations for the high firing neurons. Supplementary Fig. 6b shows that the trend of correlations of high firing neurons to performance is kept when we correlate the performance to the residuals of the firing rates ($r = 0.9123$, $p < 1e-20$), yet, slightly lower. However, we could still argue that movement during reward is not really modeled by the trajectories or the speed. For that reason, we decided to repeat the figure 3b of the main manuscript using the firing rate without the reward segment. This does not change the claim that high-firing rate neurons tend to be more significantly modulated than low firing rate ones ($p = 0.016$, chi-square test). The observation that high-firing rate neurons are inclined to be more negative correlated than positive correlated does not reach significance, but the tendency is still clear.

Reviewer #2: “Neuronal firing rates are predictive of task-performance” Why not compare the actual data correlations to the permuted data correlations, thus creating a unique empirical distribution for each ensemble? This would be more prudent analysis given the unknown nature of the distribution of predicted vs actual outcomes.

Reviewer #2: L 144 – The statement “This demonstrates that the relation between firing rate and performance is better encoded by a population of

cells” is not really backed up by more than an apples and oranges comparison. This is an interesting point and indeed recent publications have shown striking differences between single unit and ensemble information coding (Hyman et al., 2012; Ma et al., 2015; Hyman et al., 2017) but that is not really what was being compared here. An LM prediction of performance compared with actual performance is not the same as FR compared with performance. More analysis of ensemble/single unit differences is needed to support this strong a claim. That or make the claim more suggestive in nature and cite past articles showing this effect.

We agree with the reviewer that the analyses comparing the individual neurons with the population are not robust enough to hold the claims that we were making in the manuscript. In fact, as pointed out by the reviewer, it is a very interesting phenomenon that has to be investigated in more detail. This will deviate from the main claim that we want to make in the manuscript, therefore we have decided to not include this comparison as it does not have any effect on the final conclusions of the manuscript and therefore we have deleted the segment and related figures from the manuscript.

L149 - typo “remains”

Thank you for the observation. It has been corrected.

Reviewer #2: “Neuronal firing patterns in the prefrontal cortex reflect changes in rules and strategies” – what time bins were used for this analysis? Did the authors just take a single mean for the entire trial? Each behavioral segment? A series of time bins for each trial for each cell? Without this basic information it is impossible to accurately interpret the state space

We use the trial-by-trial firing rate of neurons. Meaning a single value of firing rate per trial. We divided the number of spikes by the trial time. We have now clarified this in the main text.

L 217 – use of the word “determined” in this sentence is awkward.

We have taken the comment of the reviewer into consideration. We have changed the word in the manuscript.

Reviewer #2: The authors consistently use Euclidean distance measures, when Mahalanobis distance (which also incorporates the covariance matrix) is usually used in these analyses. Why did the authors make this choice? Especially, in light of the acknowledged modeling of the analytical approach on the Durstewitz et al., 2008.

We understand the referee’s unease feeling with the use of Euclidean distance instead of Mahalanobis. The reasons behind it are: First, in this type of analyses, Euclidean has been shown to give the same results as Mahalanobis (Durstewitz et al.,2010). Second, we preferred to use a more direct measurement of distance. The Mahalanobis uses the covariance matrix to correct “distortions” of the clusters or the space, which we did not have a prior assumption of it. Third, in the Mahalanobis distance, the number of observations needs to be bigger than the number of dimensions (we need a non-singular covariance matrix, in order to invert it. The Mahalanobis distance needs the inverse of the covariance

matrix). Unfortunately, when calculating the distances from the centre of masses to the different clusters, some of these clusters will be formed by fewer observations than dimensions (for example 20 trials for a rule vs 25 neurons in the session), therefore the Mahalanobis distance cannot be computed. We wanted to compute everything in the full state-space. However, it is true that we need to demonstrate that the assumptions are still true using Mahalanobis (Which we have included in the manuscript). In summary: First, due to the fact that k-means uses Euclidean distance to cluster the data, we use a k-medoids, which is a similar version of k-means but can be used with Mahalanobis distance. The data shows a level of clustering similar to that of the k-means, and it is significantly different from the shuffle data (Supplementary Fig. 4a, $p = 1.074e-04$, Wilcoxon sign-rank test). Second, in order to analyse the contribution of the number of rules vs the number of trials to the Mahalanobis distance between trials, we have used principal component analyses as a dimensionality reduction (due to the previously described problem of the higher number of dimensions in comparison to the number of observations). After taking the three first principal components as our new dimensions (which describe the majority of the data), we re-did the GLM and obtained similar results: the Mahalanobis distance between clusters is significantly explained by the number of rules ($p = 4.7569e-05$) but not by the number of trials in between the clusters ($p=0.76405$). In addition, the interaction term is not significant ($p=0.646$). The plots of the partial correlations can be observed in the Supplementary Fig. 4b (**top**: Partial correlation of the Mahalanobis distance between clusters vs the number of trials in between while controlled by the number of rules in between; **bottom**: Partial correlation of the Mahalanobis distance between clusters vs the number of rules in between while controlled by the number of trials). Third, similar to figure 5g, we calculated if the distances between non-repeated rules are different from the distances between repeated ones. Matching with the previously presented data, there is no significant difference between them ($p = 0.248$, Supplementary figure 4c).

Reviewer #2: First paragraph of the Discussion – the authors refer to “two complementary neuronal groups” – presumably these are the neurons that are either positively or negatively correlated with behavior. These groups are not mentioned in other analyses. Were the ensembles split for the state space analyses? Is this just sloppy writing or is something not clear in the Methods?

In fact, as the referee pointed out, we have found two groups of neurons. One which firing rate is positively correlated to performance, and another for which firing rate is negatively correlated with performance. We called them complementary because one is positive and the other is negative. We only separated neurons between these two groups in order to calculate the number of high and low-firing neurons belonging to them. However, we did not split the data when analysing the state-space.

Reviewer #2: The discussion of reservoir network models is nice at the end of the Discussion section, but it fails to bring up temporally driven reservoir models like those supported by primate recordings (Bernacchia et al., 2013). Considering the authors bring these models up as a counter to time-driven network changes, this might allow for a nice discussion of how to incorporate both ideas.

We appreciate the suggestion of the reviewer, which enriched our manuscript. We have added the reference and implications to the discussion.

Reviewer #2: The “Analyses and Statistics” section of the Methods is poorly organized, choppy, and lacking. For involved layered analyses like this it generally best to have the Methods section read more narratively, while obviously still containing all the necessary details. This way, what analysis choices were made and why can be justified

We apologised for the lack of clarity in the description of our analyses. We have improved the narrative of the methods section.

REVIEWERS' COMMENTS:

Reviewer #1 (Remarks to the Author):

The authors have adequately responded to my concerns.

Figure 4: typo "Differential"

figure 6: inconsistent capitalization of T/trajectories.

Reviewer #2 (Remarks to the Author):

The authors have altered their manuscript in accordance with mine and the other reviewer's recommendations quite admirably. I find the overlapping tracks analysis to be particularly compelling, along with the speed/FR analysis. My only concern would be that the behavior and raw neural data are sufficiently represented in the main paper. Thus, I would like to see Suppl. Figs 1, 5, and 6 added to the main text.

We thank the referees for their overall positive evaluation of our manuscript and for their most helpful suggestions. We have improved the manuscript according to their comments.

Point-to-point reply to referees:

Reviewer #1 (Remarks to the Author):

The authors have adequately responded to my concerns.

Figure 4: typo "Differential"

figure 6: inconsistent capitalization of T/trajectories.

Thank you for the observation. The typo and the inconsistency have been corrected

Reviewer #2 (Remarks to the Author):

The authors have altered their manuscript in accordance with mine and the other reviewer's recommendations quite admirably. I find the overlapping tracks analysis to be particularly compelling, along with the speed/FR analysis. My only concern would be that the behavior and raw neural data are sufficiently represented in the main paper. Thus, I would like to see Suppl. Figs 1, 5, and 6 added to the main text.

We thank the reviewer for the suggestion which improves the manuscript. The suggested supplementary figures have been added to the main text.